# Marginal Zinc Deficiency Promotes Pancreatic Islet Enlargement While Zinc Supplementation Improves the Pancreatic Insulin Response in Zucker Diabetic Fatty Rats

**DOI:** 10.3390/nu16121819

**Published:** 2024-06-10

**Authors:** Leslie Rech, Peter Zahradka, Carla G. Taylor

**Affiliations:** 1Department of Food and Human Nutritional Sciences, University of Manitoba, Winnipeg, MB R3T 2N2, Canadapzahradka@sbrc.ca (P.Z.); 2Canadian Centre for Agri-Food Research in Health and Medicine, St. Boniface Albrechtsen Research Centre, 351 Tache Avenue, Winnipeg, MB R2H 2A6, Canada; 3Department of Physiology and Pathophysiology, University of Manitoba, Winnipeg, MB R3E 0J9, Canada

**Keywords:** marginal zinc deficiency, zinc supplementation, pancreatic islets, glycemic control

## Abstract

Zinc deficiency has been associated with the worsening of diabetes while zinc supplementation has been proposed to ameliorate diabetes. This study examined the effects of marginal zinc deficiency (MZD) and zinc supplementation (ZS) on obesity, glycemic control, pancreatic islets, hepatic steatosis and renal function of Zucker diabetic fatty (ZDF) rats. Male ZDF rats were fed an MZD, zinc control (ZC) or ZS diet (4, 30 and 300 mg Zn/kg diet, respectively), and lean Zucker rats were fed a ZC diet for 8 weeks. MZD and ZS did not alter body weight or whole-body composition in ZDF rats. MZD ZDF rats had reduced zinc concentrations in the femur and pancreas, a greater number of enlarged pancreatic islets and a diminished response to an oral glucose load based on a 1.8-fold greater incremental area-under-the-curve (AUC) for glucose compared to ZC ZDF. ZS ZDF rats had elevated serum, femur and pancreatic zinc concentrations, unchanged pancreatic parameters and a 50% reduction in the AUC for insulin compared to ZC ZDF rats, suggesting greater insulin sensitivity. Dietary zinc intake did not alter hepatic steatosis, creatinine clearance, or levels of proteins that contribute to insulin signaling, inflammation or zinc transport in epididymal fat. Potential adverse effects of ZS were suggested by reduced hepatic copper concentrations and elevated serum urea compared to ZC ZDF rats. In summary, ZS improved the pancreatic insulin response but not the glucose handling. In contrast, reduced zinc status in ZDF rats led to impaired glucose tolerance and a compensatory increase in the number and size of pancreatic islets which could lead to β-cell exhaustion.

## 1. Introduction

Diabetes is the most common endocrine disease worldwide, with type 2 diabetes accounting for 90–95% of diabetes cases [1]. Diabetes is characterized by chronically high fasting and postprandial blood glucose levels [2]. Dyslipidemia, especially elevated triglycerides and low HDL-cholesterol, are common in type 2 diabetes [2]. Elevated levels of circulating fatty acids can interfere with insulin signaling and enhance hyperinsulinemia as well as islet cell hyperplasia [2]. An excessive accumulation of fat in the liver, known has hepatic steatosis, is common in type 2 diabetes, and may precede or exacerbate hepatic insulin resistance [2]. Meanwhile, hyperglycemia contributes to diabetic nephropathy, resulting in renal damage and end-stage renal disease [3].

There has been considerable interest in the potential role of zinc in ameliorating metabolic disorders such as insulin resistance and type 2 diabetes given that zinc is required for the synthesis, crystallization and secretion of insulin by pancreatic β-cells, that zinc is co-secreted with insulin, and that zinc acts on α-cells to inhibit glucagon secretion [4,5,6]. In addition, zinc modulates insulin signaling while many zinc-containing metalloproteins and metalloenzymes are involved in glucose and lipid metabolism [7,8]. It has also been proposed that hyperglycemia and polyuria contribute to hyperzincuria and hypozincemia, and that urinary zinc levels can be used as an indicator of the extent of diabetes since hyperzincuria becomes exacerbated with progression to diabetes [7,9].

Zinc status may be reduced in type 2 diabetes, given reports of lower serum zinc concentrations in individuals with diabetes than those without diabetes [9]. The presence of zinc deficiency (defined as serum zinc concentrations below 10.7 μmol/L) in people with diabetes may contribute to impaired glucose-stimulated insulin secretion and various diabetes-associated complications [7]. On the other hand, zinc may have many protective effects against diabetes and its associated comorbidities [10,11]. For example, in a prospective study conducted by Sun et al. [12], women in the highest quintile for dietary zinc intake had a 20% lower risk of developing type 2 diabetes. Additionally, a meta-analysis has shown that zinc supplementation improves glycemic control for both prevention and management of diabetes [13]. Thus, the literature in this area has led to the overarching hypothesis that marginal zinc deficiency worsens and zinc supplementation improves diabetes-related parameters; however, there is a gap in understanding the in vivo effects of zinc on the pancreas versus the periphery in the context of type 2 diabetes and inter-organ metabolism.

Previous rodent studies investigating the effects of zinc deficiency or supplementation on diabetes-related parameters have been confounded by changes in body weight, and there has not been a comprehensive assessment of key organs for both positive and adverse effects. Thus, the objective of the present study was to determine the effects of marginal zinc deficiency (MZD) and zinc supplementation (ZS) for 8 weeks on obesity, glycemic control, pancreatic islets, hepatic steatosis and renal function in the Zucker diabetic fatty (ZDF) rat, a model for obesity and type 2 diabetes. Secondary objectives were to examine levels of proteins that contribute to insulin signaling, inflammation or zinc transport in epididymal fat, and to investigate the interactions among zinc status, glycemia, glucosuria and zincuria. The MZD diet (4 mg Zn/kg diet) was chosen to achieve a marginal zinc-deficient state without a reduction in body weight, whereas the level of dietary ZS (300 mg Zn/kg diet) was expected to enhance zinc status but have minimal adverse effects on the status of other divalent minerals such as copper and iron. ZDF rats have a mutation in the leptin receptor that impairs leptin signaling and results in hyperphagia, leading to weight gain, obesity, insulin resistance, hyperglycemia, dyslipidemia and hepatic steatosis [14], and thus they are a preclinical model for obesity-mediated metabolic disease and type 2 diabetes.

## 2. Methods

### 2.1. Animals and Diets

Male Zucker diabetic fatty (ZDF; ZDF-*Lepr^fa^*/*Crl*; strain code 370) rats and their lean controls (+/?; strain code 371) were obtained from Charles River Laboratories (St. Constant, PQ, Canada) at 5 weeks of age. The animals were housed singly in polycarbonate cages with bedding and stainless steel environmental enrichment items, and in a room kept at 21 °C with a 12 h light–dark cycle. The rats were acclimatized for 1 week, during which they were fed the control diet. At 6 weeks of age, 30 male ZDF rats were randomized into 3 groups and fed a marginal zinc-deficient diet (MZD, 4 mg Zn/kg diet), zinc control diet (ZC, 30 mg Zn/kg diet), or zinc-supplemented diet (ZS, 300 mg Zn/kg diet), and 7 male lean control rats were fed a ZC diet, for 8 weeks. The sample size and 8 week duration are based on the improvement of metabolic parameters with other dietary interventions in Zucker rats [15,16]. The diet formulations (Appendix A) provided 21% of energy from protein, 17% of energy from fat, and 62% of energy from carbohydrates; this is the macronutrient composition of the laboratory chow (Purina #5008) used by Charles River Laboratories to yield a type 2 diabetes phenotype in homozygous ZDF male rats [17]. Feed intake and weekly body weights were recorded. The animal care protocol was approved by the University of Manitoba Animal Care Committee and followed the guidelines of the Canadian Council on Animal Care.

### 2.2. Glycemic Control and Glucose Tolerance

At 0, 4 and 8 weeks, the rats were placed into metabolic cages (with access to water but not feed) for 5 h to obtain a urine sample for urinalysis and a saphenous blood sample (at the end of the 5 h fasting period) for assessment of glycemia and insulinemia. During week 8, an oral glucose tolerance test (OGTT) was performed to assess glucose clearance. The rats were fasted for 5 h and baseline blood samples were obtained from the saphenous vein prior to oral administration of 2 g glucose/kg (70% glucose solution). Blood samples were collected at 15, 30, 60 and 120 min after glucose consumption. Blood samples were allowed to coagulate on ice for around 30 min, and then centrifuged at 1000× *g* for 15 min at 4 °C. The serum was collected, aliquoted, frozen and stored at −80 °C for analysis of glucose and insulin. The incremental area-under-the-curve (AUC) for glucose and insulin during OGTT were calculated using the trapezoidal method [18]. The homeostasis model assessment (HOMA) of insulin resistance (IR) and beta-cell function (BCF) were also calculated [19].

### 2.3. Body Composition

During week 8, body composition was measured in vivo using an EchoMRI-700^TM^ whole-body Quantitative Magnetic Resonance (QMR) instrument (Echo Medical Systems, Houston, TX, USA). All measurements were performed in triplicate. 

### 2.4. Tissue Collection

At the end of the 8 week study, and after 12 h of fasting, the animals were euthanized using carbon dioxide asphyxiation. Body weight and body length (nose tip to anus) were recorded to calculate the Lee’s index (body weight in g^0.33^/body length in mm), a measure of adiposity [20]. Trunk blood was obtained after decapitation and centrifuged to obtain serum. Tissues were dissected, weighed, immediately frozen in liquid nitrogen and then stored at −80 °C until analyzed. A portion of the pancreas was fixed in phosphate-buffered formalin.

### 2.5. Pancreatic Islet Insulin Immunostaining and Liver Lipid Concentration

Formalin-fixed pancreas tissue was processed into paraffin blocks and sectioned (6 μm) onto positively charged slides at CancerCare Manitoba (Winnipeg, MB, Canada). Slides were deparaffinized in xylene then rehydrated in decreasing levels of ethanol-to-water solutions. Endogenous peroxidase activity was blocked using 3% H_2_O_2_ and 1× phosphate-buffered saline [137 mM NaCl, 2.7 mM KCl, 10 mM Na_2_HPO_4_, 1.8 mM KH_2_PO_4_, pH 7.4]. Slides were stained using the STAT-Q Peroxidase-DAB (3,3′-diaminobenzidine) Staining System, Insulin Mouse Monoclonal Antibody Concentrate, and Background Buster (all from Innovex Biosciences, Richmond, CA, USA). Sections were counterstained with Mayer’s hematoxylin and then dehydrated by placing the slides in increasing levels of ethanol-to-water solutions. Slides were mounted with Permount (Fisher Scientific, Saint-Laurent, QC, Canada), fixed with cover slides, and then islet cells were visualized under a microscope. Islet size was determined using the open-source image analysis software ImageJ (Version 1.50; National Institutes of Health, Bethesda, MD, USA) [21]. Total liver lipid was determined by a modified Folch method as previously described [22].

### 2.6. Serum and Urine Biochemistry

Glucose and insulin in serum from the OGTT samples, and after a 5 h fast (weeks 0, 4, 8) or 12 h fast (week 8 termination sample), were analyzed for glucose and insulin using colorimetric (Genzyme Diagnostics Glucose Reagent and Glucose Calibrator, Charlottetown, PE, Canada; SEKISUI Diagnostics DC-CAL Calibrator, Burlington, MA, USA) and electrochemiluminescence (Meso-Scale Discovery Mouse/Rat Insulin Kit, Rockville, MD, USA) kits, respectively.

End-of-study serum samples were assessed for lipid profile using a cobas c 111 clinical chemistry analyzer (Roche Diagnostics, Indianapolis, IN, USA) and for free fatty acids (FFA) using a spectrometric assay kit (Roche Diagnostics). Renal function was assessed by measuring serum and urine creatinine and urea, and urine glucose using a cobas c 111 clinical chemistry analyzer (Roche Diagnostics).

### 2.7. Mineral Status

After obtaining wet and dry weights, tissue, serum and diet samples were digested in a DigiPREP Jr. block digestion system (SCP Science, Mississauga, ON, Canada) using trace-element grade HNO_3_, and then diluted appropriately. Urine samples did not need digestion but were diluted. Inductively coupled plasma–optical emission spectrometry (ICP-OES; Varian ICP, Model-VISTA-MPX, CCD, Varian Canada, Mississauga, ON, Canada) was used to analyze minerals in the femur (zinc), liver (zinc, copper and iron), and pancreas (zinc). Atomic absorption spectroscopy (AAS; Perkin Elmer 3100 flame AAS, Perkin Elmer, Woodbridge, ON) was used for analysis of zinc in adipose tissue, diet samples, serum, and urine as AAS has a lower detection range for zinc. Quality control was monitored using bovine liver standard reference material (National Institute of Standards and Technology, Gaithersburg, MD, USA).

### 2.8. Western Immunoblotting

Protein extracts were prepared from epididymal fat and used for Western blotting following standard procedures [23]. Primary antibodies were used to explore proteins associated with energy metabolism (GPR-39, ab39227, Abcam, Waltham, MA, USA), impaired insulin signaling (suppressor of cytokine signaling-3 (SOCS-3), 2923A, Cell Signaling, Danvers, MA, USA), macrophage recruitment (monocyte chemoattractant protein (MCP-1), ab25124, Abcam), inflammation and insulin resistance (tumor necrosis factor-α (TNF-α), L-19, sc-1351, Santa Cruz, Santa Cruz, CA, USA) and zinc transport (ZnT3, G-14, sc-27508, Santa Cruz). β-Actin (4967L, Cell Signaling) was used as the loading control. Bands were scanned with a densitometer (GS 800 Imaging Densitometer, Bio-Rad Laboratories, Mississauga, ON, Canada) and the intensity for the band of interest was divided by the intensity for the loading control band.

### 2.9. Statistical Analyses

Time course and endpoint data were analyzed by repeated measures ANOVA and one-way ANOVA, respectively, followed by means testing with Duncan’s multiple range test (SAS software release 9.2, SAS Institute Inc., Cary, NC, USA). Non-normal and non-homogeneous data were analyzed following log transformation or by non-parametric testing if normality and homogeneity were not achieved by log transformation. For non-parametric testing, the Kruskal–Wallis test was followed by least significant difference post hoc testing with Tukey correction for multiple comparisons. Outliers (≥mean ± 2.5× standard deviation for the dataset) were removed before analyses. Data are reported as means ± the standard deviation (SD). Differences were considered significant at *p* < 0.05.

## 3. Results

### 3.1. Feed Intake, Body Weight and Body Composition

ZDF rats consumed 31% more feed per day and weighed 33% more than lean rats at the end of the study, and neither feed intake nor body weight were altered by diets deficient or supplemented with zinc (Table 1). Analysis of whole-body composition revealed that ZDF rats had 6-fold higher body fat and 87% less lean body tissue as well as lower total body water compared to lean rats. With respect to specific fat pads, ZDF rats had higher epididymal, peri-renal and mesenteric fat pad weights relative to body weight compared to lean rats. Despite similar body weights, visceral fat mass was reduced by 7% in MZD ZDF compared to ZS ZDF rats due to less peri-renal fat mass. Both MZD and ZS reduced the Lee’s index, a measure of adiposity based on body weight and length, compared to ZC-ZDF rats. However, protein levels of GPR-39, a zinc-sensing receptor that is activated by zinc and involved in signal transduction pathways for energy metabolism, were not altered in the epididymal fat of ZDF rats compared to lean rats or by dietary zinc level (Appendix A).

### 3.2. Zinc Status

ZS elevated serum zinc concentrations were 18–25% compared to the other groups (Figure 1A). Femur zinc concentrations were enhanced by 42% by ZS and reduced by 56% by MZD compared to ZC ZDF rats (Figure 1B) despite no differences in body weight among the ZDF rats. The ZC ZDF and lean rats had similar femur zinc concentrations. Hyperzincuria was present in ZC ZDF and ZS ZDF rats compared to ZC lean rats (Figure 1C); however, the urinary zinc excretion by MZD ZDF rats was similar to ZC lean rats, indicating a conservation of zinc when dietary zinc is limiting.

Although ZDF rats had lower liver zinc concentrations (µmol/g) than ZC lean rats, total hepatic zinc content (µg/liver) was 1.7-fold greater in ZDF rats compared to lean rats, with no differences due to dietary zinc intake (Table 2). To assess whether the level of zinc supplementation had adverse effects on other minerals, liver copper and iron were measured. Liver copper concentrations were reduced by 43% in ZS ZDF rats, and by 17–20% in MZD and ZC ZDF rats, compared to lean rats. Although total hepatic copper content was 35% lower in ZS compared to MZD and ZC ZDF rats, it was still higher than in lean rats. The ZDF groups had lower iron concentrations in the liver compared to lean rats, irrespective of dietary zinc intake, whereas the total iron content per liver was higher in ZDF rats due to the larger liver mass.

Zinc concentrations in epididymal fat tissue were not different among the groups, and the 2-fold higher zinc content in epididymal fat pads of ZDF rats paralleled the greater fat pad mass compared to lean rats. However, protein levels of the zinc transporter ZnT3 in epididymal fat were not different between ZDF and lean rats or among dietary zinc groups (Appendix A). 

### 3.3. Pancreatic Islets and Zinc

ZDF rats had a ~30% reduction in pancreas mass (absolute weight) and ~50% reduction in pancreas weight relative to body weight compared to lean rats, but there were no differences in pancreas weight due to dietary zinc intake (Table 1). Immunostaining of pancreatic sections for insulin revealed differences in islet number and size (Figure 2A). When this was quantified, MZD had a greater number of islet cells compared to ZS ZDF rats and lean rats (Figure 2B). MZD ZDF rats also had a 2-fold greater islet cell area compared to ZC ZDF rats and a 4-fold greater islet cell area compared to ZS ZDF rats (Figure 2C). The MZD ZDF rats had a pancreatic zinc concentration 30% lower than ZC ZDF rats, and ZC ZDF rats had 24% lower pancreatic concentration compared to ZC lean rats (Figure 2D). ZS did not change the number of islet cells or islet cell area compared to the ZC ZDF or lean rats; however, ZS ZDF rats had pancreatic zinc concentrations 32% greater than the ZC ZDF rats, and not different from ZC lean rats.

### 3.4. Glycemic Control

ZDF rats had elevated serum glucose concentrations compared to lean rats after a 12–14 h fast during week 8 but not during a shorter 5 h fast at weeks 0, 4 or 8 of the study (Table 3). Fasting serum glucose was not altered by dietary zinc intake. Serum insulin was elevated 10–24-fold in ZDF rats compared to lean rats throughout the study, regardless of fasting duration or dietary zinc intake. Consequently, ZDF rats had greater HOMA-IR and HOMA-BCF, indices of insulin resistance and beta-cell function, compared to lean rats throughout the study whether calculated after a 5 h or 12 h fast. ZS ZDF rats had a higher HOMA-IR compared to MZD ZDF at 4 weeks and ZC ZDF rats had a higher HOMA-BCF than MZD ZDF rats at 8 weeks after a 12 h fast.

There were no differences in serum glucose or insulin concentrations during the OGTT at week 8 (Figure 3). However, the MZD ZDF rats had a 2-fold greater AUC_glucose_ during the OGTT compared to the ZC ZDF and ZC lean groups (Table 3), indicating poorer handling of the oral glucose load. On the other hand, AUC_insulin_ was not different between MZC and ZC ZDF rats. The ZS ZDF rats had an AUC_glucose_ and AUC_insulin_ that were not different from lean ZC control rats, indicating less insulin was required to manage the glucose load compared to the other ZDF groups. The MZD ZDF and ZC ZDF rats had a 7-to-8-fold greater AUC_insulin_ compared to lean ZC rats, whereas the AUC_insulin_ of ZS ZDF rats was 3.5-fold greater than ZC lean rats. On the other hand, proteins associated with impaired insulin signaling (SOCS-3), inflammation and insulin resistance (precursor and mature TNF-α), and macrophage recruitment (MCP-1) were not altered by dietary zinc intake; however, ZDF rats had reduced SOCS-3 and precursor and mature TNF-α compared to lean rats (Appendix A).

### 3.5. Hepatic Steatosis and Circulating Lipids

ZDF rats had higher absolute and relative liver weights compared to lean rats (Table 4). In ZDF rats, ZS reduced absolute liver weight, but not liver weight relative to body weight, compared to the ZC group. ZDF rats had 2.5-fold greater hepatic lipid concentrations compared to lean rats, but there were no differences due to dietary zinc intake.

Fasting serum total cholesterol and triglyceride concentrations were elevated in ZDF rats compared to lean rats. Fasting serum LDL-cholesterol and free fatty acids were not different between ZDF and lean rats. MZD elevated fasting serum HDL-cholesterol compared to all other groups and MZD reduced triglycerides compared to ZC ZDF and ZS ZDF rats.

### 3.6. Renal Parameters

Although ZDF rats had higher absolute kidney weights compared to lean rats, kidney weight did not differ when adjusted for body weight and it was not affected by dietary zinc (Table 5). Urine output was not different between ZDF and lean rats or affected by dietary zinc intake throughout the 8 week study. At baseline, 6-week-old ZDF rats were excreting an average of 19.9 ± 5.3 nmol Zn/5 h in their urine, and over the 8 week study the hyperzincuria increased 2.3-fold in ZC ZDF rats to 44.0 ± 7.1 nmol Zn/5 h. During week 8, MZD ZDF rats had ~80% less urinary zinc excretion (absolute, and relative to creatinine or body weight) compared to ZC and ZS ZDF rats, and the urinary zinc excretion of ZC lean rats was <10% of ZC ZDF rats. 

ZC ZDF rats had greater urinary glucose and urea excretion (absolute and relative to creatinine) compared to ZC lean rats. ZS enhanced urinary glucose excretion (absolute amount and relative to creatinine) compared to all other groups. Furthermore, urinary glucose and zinc excretion were moderately correlated (r = 0.637, *p* = 0.001 and r = 0.648, *p* = 0.001 for absolute amount and relative to creatinine, respectively). Urinary urea excretion (absolute and relative to creatinine) was similar among the ZDF groups regardless of dietary zinc intake, and elevated in ZDF rats compared to ZC lean rats. 

ZS ZDF rats had elevated serum urea compared to MZD ZDF and lean rats, and lower serum creatinine compared to lean rats. This resulted in the ZS ZDF group having a higher serum urea/creatinine ratio compared to MZD ZDF and ZC lean rats. ZC ZDF rats also had an elevated urea/creatinine ratio compared to ZC lean rats. However, creatinine clearance was not different between ZDF and lean rats or among dietary zinc groups. 

## 4. Discussion

The key finding of the present study is that marginal zinc deficiency, characterized by reduced pancreatic and femur zinc concentrations but no changes in serum zinc or body weight, is sufficient to exacerbate pancreatic β-cell hypertrophy and hyperplasia 2- to 4-fold in ZDF rats. Marginal zinc deficiency in ZDF rats resulted in poorer handling of an oral glucose load as revealed by a 1.8-fold increased AUC_glucose_ during the OGTT, and reduced pancreatic function as indicated by a 31% lower HOMA-BCF (after the 12–14 h fast during week 8) compared to ZC ZDF rats. On the other hand, zinc-supplemented ZDF rats had an elevated zinc status as indicated by increased serum, femur and pancreas zinc concentration, and a reduction in insulin response (AUC_insulin_) by 50% after the oral glucose challenge, despite no alterations in pancreatic islet size or islet number compared to ZC ZDF rats. Furthermore, this study illustrates the narrow window for zinc supplementation without adverse effects on other divalent metal ions given that ZS ZDF rats had liver copper concentrations reduced by 27% compared to ZC ZDF rats. Interestingly, glucosuria was greatest in ZS ZDF rats even though their fasting and post-prandial glycemia were similar to ZC ZDF rats, and this may be related to the positive correlation observed between zincuria and glucosuria. 

The findings indicate that the pancreas is highly susceptible to marginal zinc deficiency and that this impacts the ability to handle a glucose load based on the elevated AUC_glucose_. Although there has been considerable interest in the role of pancreatic zinc and zinc transporter proteins in relation to zinc and insulin synthesis, storage and secretion from the pancreas [7,8,24,25,26], the present study is the first to visualize the effects of dietary zinc deficiency and supplementation on pancreatic islet size and number. In ZDF rats, a 30% reduction in pancreatic zinc by MZD was linked to pancreatic islet hypertrophy and hyperplasia and impaired β-cell function (elevated HOMA-BCF after the 12–14 h fast during week 8), and consequently, impaired oral glucose tolerance as reflected by increased AUC_glucose_ during week 8 compared to both ZC ZDF and ZC lean rats. Despite zinc having important roles in the synthesis and release of insulin from the pancreas, the MZD and ZC ZDF rats had similar fasting hyperinsulinemia and AUC_insulin_, indicating that the pancreatic islets of the MZD ZDF rats were producing sufficient insulin for a compensatory response to the glucose challenge and had not yet reached β-cell exhaustion. Preservation of insulin synthesis and islet function is in agreement with another study demonstrating that zinc ion staining was maintained in the islets of Langerhans, but not in the acinar cells of the exocrine pancreas, when Wistar rats were fed a zinc-deficient diet for 4 weeks [27]. The link between lower zinc status in humans and diabetes is based on studies linking reduced dietary zinc intake or serum zinc concentrations with risk for diabetes or presence of disease [7,9,28]. In the present study, the ZDF rats with marginal zinc deficiency did not have hypozincemia despite a 50% lower femur zinc concentration and a 30% lower pancreatic zinc concentration compared to both ZC ZDF and ZC lean rats. It has been notoriously difficult to assess mild zinc deficiency in humans based on circulating zinc concentrations [29,30,31] and the present study raises questions about whether the availability of a method (or a biomarker) to non-invasively assess femur or pancreatic zinc concentrations in humans would help advance our understanding of how reduced zinc status impacts the risk and/or progression of type 2 diabetes in humans.

The findings also indicate that zinc supplementation increases zinc in the pancreas and reduces the AUC_insulin_ during an oral glucose challenge even though fasting hyperinsulinemia and pancreatic islet cell size and islet number were unchanged compared to ZC ZDF rats. Notably, the pancreatic zinc concentration of the ZS ZDF group was elevated to a level similar to ZC lean rats. We have previously shown that zinc supplementation (300 mg Zn/kg diet for 6 weeks) increases pancreatic zinc of *db*/*db* mice to levels similar to lean control mice and attenuates hyperglycemia and hyperinsulinemia of zinc-supplemented *db*/*db* mice compared to zinc-deficient and zinc-adequate *db*/*db* mice [32]. In the present study, zinc supplementation reduced the AUC_insulin_ by 50% compared to ZC ZDF rats but the AUC_insulin_ was still elevated 4-fold compared the ZC lean rats. Zn supplementation improved the insulin response (AUC_insulin_) during the oral glucose challenge, but not glucose handling (AUC_glucose_), hyperglycemia, hyperinsulinemia or islet function (HOMA-BCF). Thus, zinc supplementation was effective during the post-prandial response since less insulin was required by the ZS ZDF rats for handling the glucose load, even though their AUC_glucose_ did not differ from the ZC ZDF rats. Furthermore, zinc supplementation did not affect peripheral insulin sensitivity as indicated by the HOMA-IR being similar among all three ZDF groups. Similarly, others have reported that zinc supplementation (1000 mg Zn/kg diet for 4 weeks) of *ob*/*ob* mice attenuates pancreatic insulin secretion in vivo and in vitro without improving tolerance to a glucose load or altering peripheral insulin sensitivity [33]. In the present study, proteins associated with impaired insulin signaling (SOCS-3), inflammation and insulin resistance (TNF-α) were unchanged in epididymal fat, also supporting a lack of effect of zinc on the peripheral actions of insulin. An important distinction in the literature is that the evidence for zinc improving peripheral insulin sensitivity is largely based on studies with cultured cells showing insulin-like effects of zinc on glucose transport mediated via the insulin receptor and its signaling pathway [34,35,36]. Future studies are needed to further delineate how dietary zinc affects insulin action in vivo with respect to the pancreas and the periphery.

Interestingly, zinc supplementation substantially increased glucosuria (both absolute amount and relative to creatinine) in ZDF rats despite no evidence of renal dysfunction based on kidney weights, creatinine clearance, or serum and urinary creatinine or urea. Furthermore, urinary glucose and zinc excretion were positively correlated. This association of hyperglycemia and hyperzincuria requires further investigation with respect to the interactions between zinc and glucose during glucose reabsorption by the sodium-dependent glucose cotransporter (SGLT) in renal tubule cells as it has been reported that zinc inhibits glucose transport via SGLT in pig intestinal brush border membrane vesicles [37]. Those data indicated that zinc reduces glucose affinity, but not sodium binding, to the transporter [37]. Furthermore, the zinc-finger transcription factor Sp1 regulates expression of SGLT [38], yet the effects of zinc status and diabetes on this regulatory mechanism for SGLT and the implications for glucose reabsorption have not been investigated despites studies showing the importance of zinc for Sp1 and SGLT levels in the context of cadmium toxicity and glucosuria [39,40]. Likewise, the effects of hyperglycemia on zinc reabsorption in the kidney via zinc transporters have not been addressed. In the past, it has been inferred that lower body contents of zinc and high urinary zinc excretion in diabetes are due to hyperglycemia and its interference with zinc reabsorption by renal cells [28]. However, the ZDF rats with marginal zinc deficiency in the present study had ~5-fold less urinary zinc excretion compared to ZC ZDF rats, despite a worsened response to an oral glucose load (1.8-fold greater AUC for glucose). Under these conditions, it appears that the reduced urinary zinc excretion of MZD ZDF rats was in part an adaptation to the lowered zinc intake. Meanwhile, urinary zinc excretion was unchanged in ZS ZDF rats compared to ZC ZDF rats but femur zinc concentrations were elevated ~50%, indicating that with high dietary zinc intake there is greater storage of zinc to maintain zinc homeostasis.

Femur zinc concentrations were not different between ZDF rats and lean rats fed the ZC diet, indicating no effect of obesity and diabetes on bone zinc status. However, the present study also reinforces that there is a very narrow range for zinc supplementation without negative effects on other divalent metal ions [29], as the ZS ZDF rats had a reduction in liver copper concentration and total liver copper compared to ZC ZDF rats. The negative relationship between high dietary zinc intake and copper status is well documented in rodents [41] and reflects competition in the gut for absorption. As a component of metalloenzymes, particularly oxidases, copper has important functions in redox reactions, energy metabolism, synthesis of connective tissue and neurotransmitter metabolism, as well as iron absorption and metabolism [42]. High zinc intakes are of concern for humans, with potential impacts for reduced copper status, altered iron metabolism, suppression of the immune response, and decreased HDL-cholesterol [42]. Thus, the upper limit for zinc in the Dietary Reference Intakes is set at 40 mg/day for adults compared to the Recommended Dietary Allowance of 11 mg/d for men and 8 mg/d for women [42]. If zinc supplementation is being considered for individuals with type 2 diabetes, it should be directed to those with deficient or suboptimal zinc status, personalized for the dose and duration, and include monitoring of parameters related to zinc, copper and iron status.

With respect to obesity, Zn supplementation elevated visceral fat mass (g/100 g bwt) in ZDF rats due to more peri-renal fat mass compared to MZD ZDF rats, despite both groups having similar body weights and comparable whole-body fat mass and lean mass. Given that Zn can stimulate adipogenesis and is involved in adipocyte metabolism [34,43], it is possible to speculate that marginal zinc deficiency may reduce and zinc supplementation may enhance adipogenesis and adipocyte differentiation in growing ZDF rats; however, this hypothesis needs further testing in vivo. In the present study, there were no differences in protein levels of ZnT3 (a Zn transporter) or GPR-39 (a zinc sensing receptor involved in signal transduction pathways for energy metabolism) in epididymal fat. Perhaps this lack of change fits with the observation that adipose Zn was maintained despite the differences in dietary Zn intake and serum Zn concentrations among the three ZDF groups. We analyzed adipose Zn concentrations in the present study as obesity in humans has been associated with lower serum Zn [44], and in diet-induced obese mice we had observed reduced adipose Zn [45]. Obesity is closely linked with the presence of metabolic (non-alcoholic) fatty liver disease [46]. Serum zinc concentrations are reduced in patients with non-alcoholic fatty liver disease [47] and there may be an association of lower serum zinc and greater severity of non-alcoholic fatty liver disease in humans [48,49]. However, in the present study, neither marginal Zn deficiency nor Zn supplementation altered the relative liver weight or the hepatic triglyceride levels of ZDF rats, and similarly others have reported that zinc supplementation did not alter the hepatic steatosis score of mice fed high-fat diets [50]. Dyslipidemia, especially low HDL-cholesterol and elevated triglycerides, is a common feature of type 2 diabetes in humans [2]. In the present study, marginal zinc deficiency reduced fasting serum triglycerides by 50% and elevated HDL-cholesterol by 1.8-fold without changing total cholesterol or LDL-cholesterol levels compared to ZC and ZS ZDF rats; however, it is noted that rats are not a good model for blood lipids and their metabolism in humans. Nonetheless, two meta-analyses of randomized controlled trials investigating zinc supplementation and the lipid profile of individuals with prediabetes or type 2 diabetes have reported reductions in serum triglycerides and total cholesterol, and elevations in HDL-cholesterol [51,52].

The present study compared length of fasting (5 h versus 12–14 h during week 8), given that animal ethics committees are requiring experimental data to justify longer periods of fasting (e.g., overnight). The longer fasting period was required for hyperglycemia in ZDF rats and to reveal the dysregulation of hepatic gluconeogenesis. This was also observed to a lesser degree in the lean (+/?) Zucker rats. On the other hand, circulating insulin concentrations were lower after the longer fasting period. Thus, the combination of changes in glucose and insulin values after different lengths of fasting impacted the calculations for insulin resistance (HOMA-IR) and β-cell function (HOMA-BCF). These data reinforce the importance of documenting the length of fasting state for the interpretation of the biological response and for comparing results among different publications.

## 5. Conclusions

The main effect of marginal zinc deficiency was impaired pancreatic function and reduced oral glucose tolerance of ZDF rats, while zinc supplementation reduced the amount of insulin required for disposal of an oral glucose load, an indication of improved insulin sensitivity. Otherwise, marginal zinc deficiency and zinc supplementation had little effect on obesity, hepatic steatosis or other indicators of metabolic function in ZDF rats. Marginal zinc deficiency reduced urinary zinc excretion, zinc storage in the femur and tissues levels in pancreas to compensate for reduced dietary intake. Meanwhile, zinc supplementation elevated serum, femur and pancreas zinc concentrations, but it also reduced liver copper by 30%. Although zinc-supplemented ZDF rats had similar fasting glycemia and urinary zinc excretion as ZC ZDF rats, glucosuria was greater with zinc supplementation, supporting the view that glucosuria and hyperzincuria are linked. In conclusion, adequate zinc intake and adequate zinc status are essential for maintaining pancreatic function, glucose tolerance and insulin sensitivity, but at the same time, there is a small window for which the potential benefits of zinc supplementation do not have negative impacts on zinc metabolism or other trace minerals.

## Figures and Tables

**Figure 1 nutrients-16-01819-f001:**
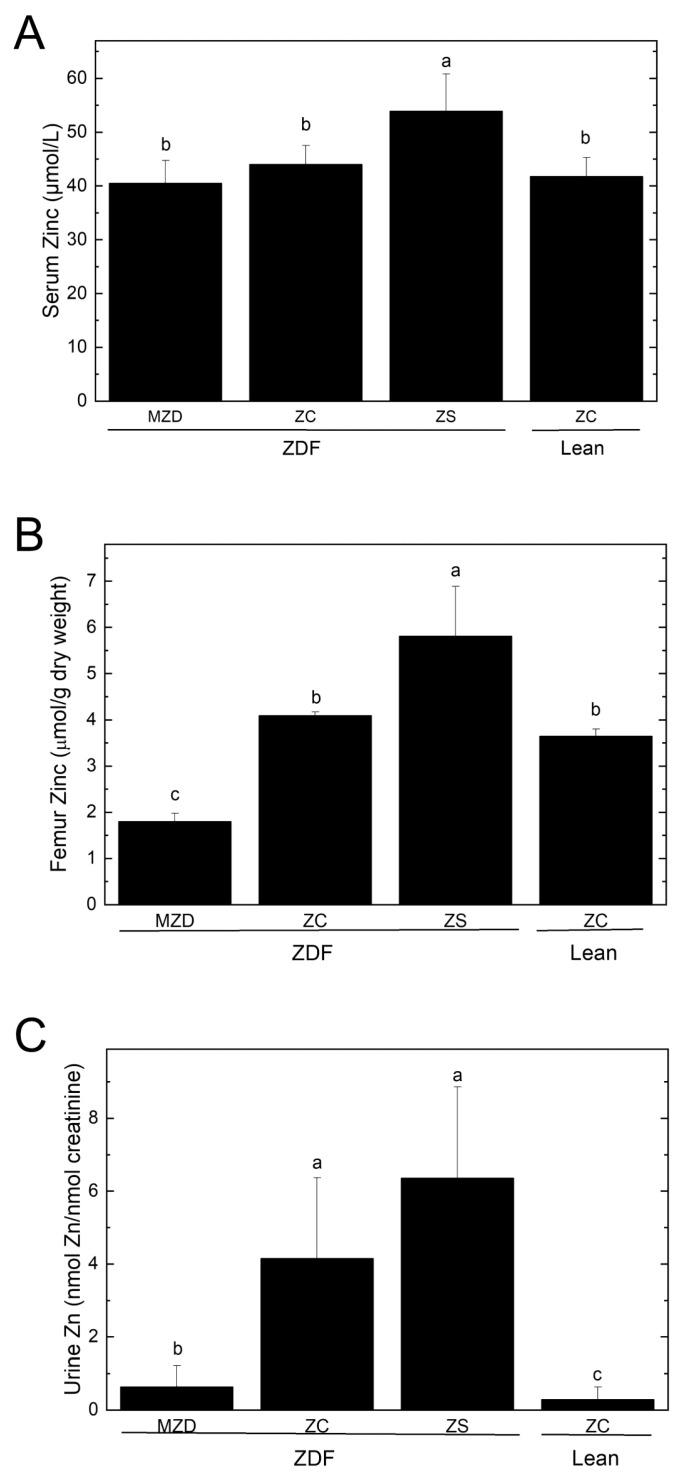
Serum (**A**) and femur (**B**) zinc concentrations, and urine zinc excretion (**C**). Values are means ± SD, *n* = 6–10. Different superscript letters indicate significant differences (*p* < 0.05) among means. Abbreviations: Lean, lean (+/?) control rats; MZD, marginal zinc-deficient diet; ZC, zinc control diet; ZDF, Zucker diabetic fatty rats; and ZS, zinc-supplemented diet.

**Figure 2 nutrients-16-01819-f002:**
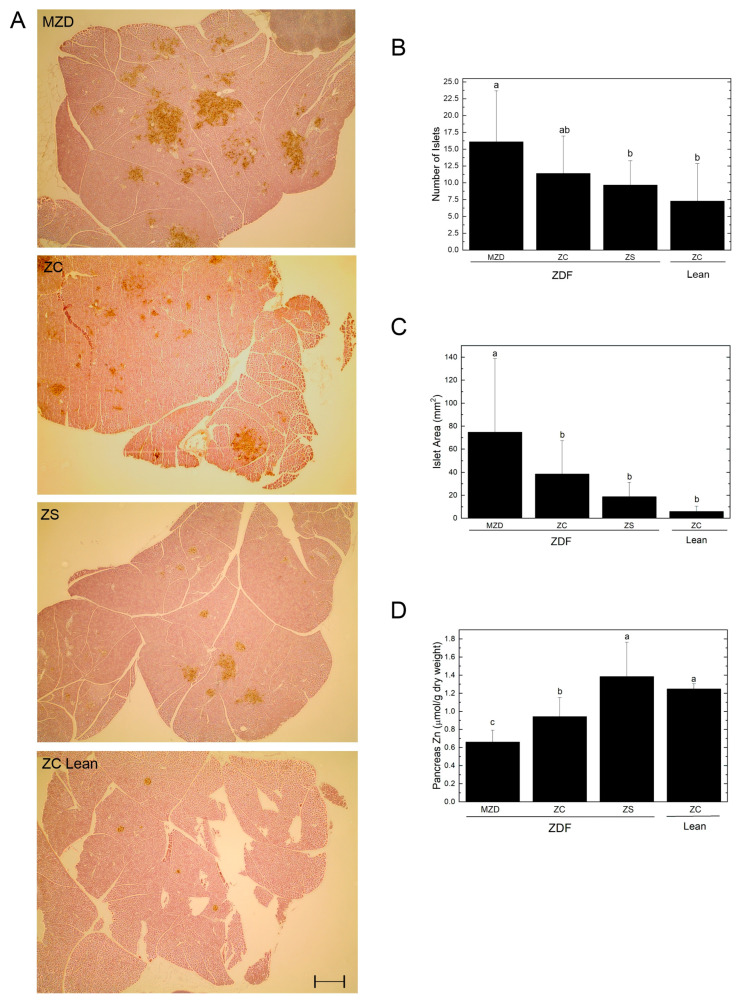
Pancreatic islet insulin immunostaining (**A**), number of islet cells (**B**), islet cell area (**C**) and pancreatic zinc concentrations (**D**). Representative images show pancreatic islets identified by insulin immunostaining and visualized by DAB in brown. Scale bar = 0.5 mm. Values are means ± SD, *n* = 10 except *n* = 9 for ZS ZDF and *n* = 7 for ZC Lean. Different superscript letters indicate significant differences (*p* < 0.05) among means. Abbreviations: DAB, 3,3′-diaminobenzidine; Lean, lean (+/?) control rats; MZD, marginal zinc-deficient diet; ZC, zinc control diet; ZDF, Zucker diabetic fatty rats; and ZS, zinc-supplemented diet.

**Figure 3 nutrients-16-01819-f003:**
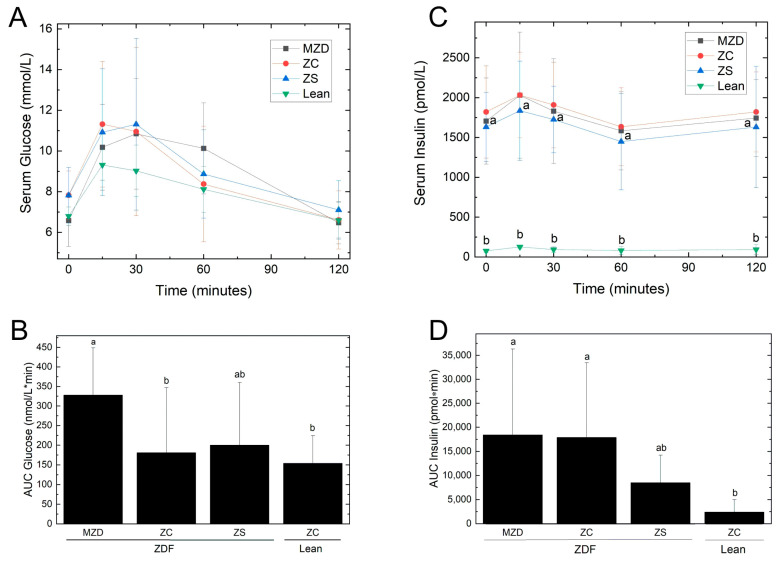
Serum glucose (**A**) and insulin (**C**) concentrations and AUC for glucose (**B**) and insulin (**D**) during an oral glucose tolerance test. Values are means ± SD for changes from baseline, *n* = 10 except *n* = 9 for ZS ZDF and *n* = 7 for ZC Lean. There were significant main effects of time but not time × diet interaction for serum glucose and insulin. Different superscript letters indicate significant differences (*p* < 0.05) among means for AUC for glucose and insulin. Abbreviations: AUC, area under the curve; Lean, lean (+/?) control rats; MZD, marginal zinc-deficient diet; ZC, zinc control diet; ZDF, Zucker diabetic fatty rats; and ZS, zinc-supplemented diet.

**Table 1 nutrients-16-01819-t001:** Feed intake, body weight, body composition and pancreas weight.

	MZD ZDF	ZC ZDF	ZS ZDF	ZC Lean
Daily Feed Intake (g/d)	26.6 ± 4.5 ^a^	28.4 ± 5.4 ^a^	27.7 ± 5.8 ^a^	21.1 ± 3.8 ^b^
Final Body Weight (g)	425 ± 16 ^a^	424 ± 23 ^a^	420 ± 20 ^a^	321 ± 5 ^b^
Whole-Body Composition ^1^				
Fat Mass (g)	164 ± 6 ^a^	168 ± 10 ^a^	169 ± 10 ^a^	28 ± 4 ^b^
Lean Mass (g)	247 ± 19 ^a^	239 ± 21 ^a^	242 ± 12 ^a^	280 ± 12 ^b^
Free Water (g)	1.36 ± 0.65	1.31 ± 0.26	1.09 ± 0.50	1.00 ± 0.21
Total Water (g)	209 ± 16 ^a^	202 ± 18 ^a^	205 ± 10 ^a^	235 ± 12 ^b^
Fat Pads				
Epididymal Fat (g/100 g bwt)	2.16 ± 0.18 ^a^	2.18 ± 0.16 ^a^	2.21 ± 0.15 ^a^	0.89 ± 0.12 ^b^
Peri-renal Fat (g/100 g bwt)	3.46 ± 0.17 ^a^	3.64 ± 0.24 ^b^	3.80 ± 0.19 ^b^	1.30 ± 0.14 ^c^
Mesenteric Fat (g/100 g bwt)	1.32 ± 0.11 ^a^	1.35 ± 0.18 ^a^	1.42 ± 0.18 ^a^	0.56 ± 0.07 ^b^
Visceral Fat ^2^ (g/100 g bwt)	6.94 ± 0.31 ^b^	7.17 ± 0.27 ^ab^	7.43 ± 0.37 ^a^	2.76 ± 0.26 ^c^
Lee’s Index (g^0.33^/mm)	0.0345 ± 0.0004 ^b^	0.0350 ± 0.0004 ^a^	0.0345 ± 0.0003 ^b^	0.0310 ± 0.0005 ^c^
Pancreas Weight				
(g)	0.99 ± 0.16 ^b^	0.90 ± 0.18 ^b^	1.03 ± 0.16 ^b^	1.46 ± 0.12 ^a^
(g/100 g bwt)	0.234 ± 0.036 ^b^	0.214 ± 0.049 ^b^	0.245 ± 0.035 ^b^	0.457 ± 0.041 ^a^

^1^ Body composition determined in vivo by whole-body Quantitative Magnetic Resonance during week 8. ^2^ Sum of epididymal, peri-renal and mesenteric fat pads. Values are means ± SD, *n* = 10 except *n* = 9 for ZS ZDF rats and *n* = 7 for ZC Lean. Different superscript letters within a row indicate significant differences (*p* < 0.05) among means. Abbreviations: bwt, body weight; Lean, lean (+/?) control rats; MZD, marginal zinc-deficient diet; ZC, zinc control diet; ZDF, Zucker diabetic fatty rats; and ZS, zinc-supplemented diet.

**Table 2 nutrients-16-01819-t002:** Hepatic minerals and epididymal fat zinc.

	MZD ZDF	ZC ZDF	ZS ZDF	ZC Lean
Liver				
Zinc Concentration (µmol/g dry weight)	1.02 ± 0.11 ^b^	1.01 ± 0.15 ^b^	1.06 ± 0.14 ^b^	1.47 ± 0.11 ^a^
Total Zinc Content (µmol/liver)	6.64 ± 0.77 ^a^	7.34 ± 0.65 ^a^	6.81 ± 0.90 ^a^	4.11 ± 0.0.30 ^b^
Copper Concentration (µmol/g dry weight)	0.215 ± 0.034 ^b^	0.203 ± 0.032 ^b^	0.148 ± 0.029 ^c^	0.259 ± 0.047 ^a^
Total Copper Content (µmol/liver)	1.397 ± 0.144 ^a^	1.484 ± 0.210 ^a^	0.947 ± 0.164 ^b^	0.727 ± 0.119 ^c^
Iron Concentration (µmol/g dry weight)	9.66 ± 2.68 ^ab^	7.61 ± 2.18 ^b^	8.25 ± 2.74 ^b^	11.00 ± 1.59 ^a^
Total Iron Content (µmol/liver)	64.3 ± 23.6 ^a^	54.3 ± 9.4 ^a^	53.7 ± 20.2 ^a^	30.9 ± 4.6 ^b^
Epididymal Fat				
Zinc Concentration (μmol/g dry weight)	0.190 ± 0.139	0.127 ± 0.062	0.148 ± 0.100	0.086 ± 0.024
Total Zinc (µg/epididymal fat)	13.4 ± 1.9 ^a^	13.3 ± 3.7 ^a^	13.4 ± 4.2 ^a^	4.5 ± 1.4 ^b^

Values are means ± SD, *n* = 6–10/group. Different superscript letters indicate significant differences (*p* < 0.05) among means. Abbreviations: Lean, lean (+/?) control rats; MZD, marginal zinc-deficient diet; ZC, zinc control diet; ZDF, Zucker Diabetic Fatty rats; ZS, zinc-supplemented diet.

**Table 3 nutrients-16-01819-t003:** Serum glucose and insulin, HOMA-IR and pancreatic β-cell function.

	MZD ZDF	ZC ZDF	ZS ZDF	ZC Lean
A. After a 5 h fast during the study				
Glucose (mmol/L)				
Week 0	7.14 ± 0.47	7.27 ± 0.64	7.08 ± 0.55	6.49 ± 0.45
Week 4	6.53 ± 0.69	7.06 ± 1.31	7.31 ± 1.42	6.13 ± 0.36
Week 8	6.58 ± 1.27	7.83 ± 1.19	7.83 ± 1.37	6.80 ± 0.46
Insulin (pmol/L)				
Week 0	434 ± 165 ^a^	456 ± 104 ^a^	383 ± 121 ^a^	43 ± 7 ^b^
Week 4	1195 ± 533 ^a^	1670 ± 657 ^a^	1830 ± 740 ^a^	68 ± 14 ^b^
Week 8	1706 ± 541 ^a^	1820 ± 579 ^a^	1630 ± 436 ^a^	76 ± 20 ^b^
HOMA-IR (µU/mL × mmol/L)				
Week 0	20.2 ± 8.6 ^a^	21.6 ± 5.7 ^a^	17.3 ± 5.1 ^a^	1.8 ± 0.3 ^b^
Week 4	50.4 ± 21.5 ^b^	66.7 ± 24.9 ^ab^	87.7 ± 45.2 ^a^	2.7 ± 0.6 ^c^
Week 8	72.5 ± 30.09.5 ^a^	91.7 ± 33.8 ^a^	84.9 ± 38.6 ^a^	3.3 ± 0.8 ^b^
HOMA-BCF (mU/mmol)				
Week 0	341 ± 111 ^a^	350 ± 76 ^a^	319 ± 121 ^a^	42 ± 10 ^b^
Week 4	1185 ± 672 ^a^	1450 ± 573 ^a^	1495 ± 696 ^a^	76 ± 20 ^b^
Week 8	1535 ± 669 ^a^	1283 ± 464 ^a^	1128 ± 269 ^a^	68 ± 23 ^b^
B. After a 12–14 h fast at the end of week 8				
Glucose (mmol/L)	17.5 ± 1.8 ^a^	15.8 ± 3.3 ^a^	16.5 ± 2.0 ^a^	12.5 ± 1.6 ^b^
Insulin (pmol/L)	888 ± 201 ^a^	1055 ± 307 ^a^	920 ± 256 ^a^	46 ± 24 ^b^
HOMA-IR (µU/mL × mmol/L)	112 ± 37 ^a^	108 ± 37 ^a^	99 ± 33 ^a^	5 ± 3 ^b^
HOMA-BCF (mU/mmol)	180 ± 48 ^b^	261 ± 96 ^a^	206 ± 57 ^ab^	13 ± 7 ^c^

Values are means ± SD, *n* = 6–10. Different superscript letters within a row indicate significant differences (*p* < 0.05) among means; an absence of letters indicates no significant differences. Abbreviations: AUC, area under the curve; HOMA-BCF, homeostasis model of assessment—beta-cell function; HOMA-IR, homeostasis model of assessment—insulin resistance; Lean, lean (+/?) control rats; MZD, marginal zinc-deficient diet; ZC, zinc control diet; ZDF, Zucker diabetic fatty rats; and ZS, zinc-supplemented diet.

**Table 4 nutrients-16-01819-t004:** Hepatic steatosis and lipidemia.

	MZD ZDF	ZC ZDF	ZS ZDF	ZC Lean
Liver (g)	20.6 ± 2.2 ^ab^	22.4 ± 2.8 ^a^	20.0 ± 1.6 ^b^	10.6 ± 0.4 ^c^
Liver (g/100 g bwt)	4.88 ± 0.48 ^a^	5.31 ± 0.75 ^a^	4.78 ± 0.44 ^a^	3.32 ± 0.15 ^b^
Liver Lipid Concentration (mg lipid/g tissue)	78.1 ± 29.3 ^a^	97.2 ± 33.7 ^a^	80.1 ± 24.8 ^a^	34.9 ± 5.7 ^b^
Serum Total Cholesterol (mmol/L)	5.89 ± 1.00 ^a^	6.25 ± 0.79 ^a^	6.04 ± 1.19 ^a^	2.65 ± 0.17 ^b^
Serum LDL-C (mmol/L)	0.424 ± 0.283	0.200 ± 0.172	0.362 ± 0.385	0.424 ± 0.065
Serum HDL-C (mmol/L)	3.78 ± 0.67 ^a^	2.08 ± 1.187 ^b^	2.20 ± 0.60 ^b^	2.38 ± 0.13 ^b^
Serum TG (mmol/L)	7.0 ± 2.2 ^b^	13.5 ± 4.8 ^a^	12.2 ± 3.6 ^a^	0.94 ± 0.28 ^c^
Serum FFA (mmol/L)	0.329 ± 0.061	0.317 ± 0.084	0.340 ± 0.057	0.283 ± 0.035

Values are means ± SD, *n* = 10 except *n* = 9 for ZS ZDF and *n* = 7 for ZC Lean. Different superscript letters within a row indicate significant differences (*p* < 0.05) among means. Abbreviations: bwt, body weight; FFAs, free fatty acids; HDL-C, high density lipoprotein-cholesterol; LDL-C, low density lipoprotein-cholesterol; Lean, lean (+/?) control rats; MZD, marginal zinc-deficient diet; TGs, triglycerides; ZC, zinc control diet; ZDF, Zucker diabetic fatty rats; and ZS, zinc-supplemented diet.

**Table 5 nutrients-16-01819-t005:** Renal parameters.

	MZD ZDF	ZC ZDF	ZS ZDF	ZC Lean
Kidney Weights				
Left Kidney (g)	1.47 ± 0.14 ^a^	1.45 ± 0.09 ^a^	1.47 ± 0.18 ^a^	1.24 ± 0.08 ^b^
Left Kidney (g/100 g bwt)	0.349 ± 0.035	0.345 ± 0.038	0.352 ± 0.040	0.388 ± 0.023
Right Kidney (g)	1.47 ± 0.14 ^a^	1.45 ± 0.09 ^a^	1.47 ± 0.18 ^a^	1.24 ± 0.08 ^b^
Right Kidney (g/100 g bwt)	0.361 ± 0.036	0.374 ± 0.044	0.382 ± 0.007	0.394 ± 0.002
Urine Output (mL/5 h)				
Week 0	3.20 ± 0.76	3.06 ± 0.84	3.43 ± 1.25	1.67 ± 0.74
Week 4	3.23 ± 1.05	4.02 ± 3.00	2.73 ± 1.24	2.76 ± 1.09
Week 8	3.23 ± 1.55	3.96 ± 1.59	3.51 ± 1.55	3.67 ± 1.47
Urinanalysis (Week 8)				
Creatinine (nmol/5 h)	12.1 ± 2.9	10.8 ± 2.5	10.0 ± 3.4	12.9 ± 2.8
Zinc (nmol/5 h)	7.8 ± 6.6 ^b^	44.0 ± 22.5 ^a^	64.2 ± 29.2 ^a^	3.5 ± 3.2 ^c^
Zinc (nmol/100 g bwt)	1.8 ± 1.5 ^b^	10.4 ± 5.3 ^a^	15.3 ± 7.0 ^a^	1.1 ±1.0 ^b^
Glucose (nmol/5 h)	7.8 ± 9.2 ^b^	6.4 ± 2.9 ^b^	31.1 ± 24.1 ^a^	1.8 ± 1.1 ^c^
Glucose/Creatinine (µmol/nmol)	1757 ± 3042 ^b^	597 ± 325 ^b^	3064 ± 2318 ^a^	136 ± 61 ^c^
Urea (mmol/5 h)	2.1 ± 0.4 ^a^	2.0 ± 0.5 ^a^	2.0 ± 0.7 ^a^	1.3 ± 0.4 ^b^
Urea/Creatinine (mmol/mmol)	185 ± 19 ^a^	184 ± 31 ^a^	196 ± 41 ^a^	96 ± 12 ^b^
Serum Biochemistry (Week 8)				
Urea (µmol/L)	3.89 ± 0.40 ^b^	4.07 ± 0.59 ^ab^	4.52 ± 0.72 ^a^	3.68 ± 0.45 ^b^
Creatinine (mmol/L)	29.1 ± 3.7 ^ab^	25.8 ± 5.7 ^b^	26.7 ± 4.3 ^b^	32.9 ± 2.3 ^a^
Urea:Creatinine (mmol/L:mmol/L)	136 ± 24 ^bc^	169 ± 62 ^ab^	173 ± 38 ^a^	112 ± 10 ^c^
Creatinine Clearance (Week 8)				
Creatinine Clearance (mL/min)	1.39 ± 0.41	1.33 ± 0.39	1.27 ± 0.42	1.32 ± 0.32

Values are means ± SD, *n* = 7–10. Different superscript letters within a row indicate significant differences (*p* < 0.05) among means; an absence of letters indicates no significant differences. Abbreviations: Lean, lean (+/?) control rats; MZD, marginal zinc-deficient diet; ZC, zinc control diet; ZDF, Zucker diabetic fatty rats; and ZS, zinc-supplemented diet.

## Data Availability

The data supporting the conclusions of this article will be made available by the authors on request and according to institutional procedures.

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
