# Peer review of "Marginal Zinc Deficiency Promotes Pancreatic Islet Enlargement While Zinc Supplementation Improves the Pancreatic Insulin Response in Zucker Diabetic Fatty Rats"

_nutrients, 2024, doi:10.3390/nu16121819_

Round 1

Reviewer 1 Report

Comments and Suggestions for Authors

1. Were the potential effects of gender differences on experimental results considered in the study design?

2. Sample size and representativeness: The article mentioned that 30 male ZDF rats were used in the experiment, but did not mention whether the sample size was calculated in advance.

3. Why was the experiment period chosen to be eight weeks? Is there any theoretical basis for the above?

4. Dosage of zinc supplementation: The zinc supplementation dose used in the study (300 mg Zn/kg diet) varied significantly from the zinc control dose (30 mg Zn/kg diet) to the marginal deficiency dose (4 mg Zn/kg diet). Consideration needs to be given to whether such high-dose supplementation is clinically necessary or whether there is a risk of excessive intake. How to determine the best dose supplement?

5. In the selection of zinc supplement dosage, have you considered the effects of long-term supplementation on animal health?

6. ZDF rats with borderline zinc deficiency (MZD) reported reduced zinc concentrations in the femur and pancreas, but no changes in serum zinc or body weight. This may suggest that assessing zinc status by serum zinc concentration alone may not be accurate enough. More biomarkers need to be considered to more fully assess zinc nutritional status.

7. The potential role of zinc in diabetes management is discussed, but its specific guiding implications for clinical practice, including recommended dosage and monitoring requirements for zinc supplementation, are not specified.

Comments on the Quality of English Language

The use of professional terms is basically correct, and the language is clear and fluent. It should be noted that informal and subjective language should be avoided to ensure the authority of the paper.

Author Response

Thank you for the review of our manuscript and your insightful comments for improving our manuscript.

  1. Were the potential effects of gender differences on experimental results considered in the study design?

Response:  It would be interesting to investigate sex differences in this model, however, the budget did not allow us to purchase twice the number of ZDF rats and complete twice the number of analyses. Thus, we completed the study with males since they are the most frequently used sex for such experiments.

  1. Sample size and representativeness: The article mentioned that 30 male ZDF rats were used in the experiment, but did not mention whether the sample size was calculated in advance.

Response:  The sample size of n=10 ZDF rats per group was based on our experience with conducting similar assessments for oral glucose tolerance, body composition, pancreatic islets, hepatic steatosis, etc., in fa/fa Zucker rats with other types of dietary interventions (e.g. conjugated linoleic acid, high-protein diets) and detecting significant differences [1-5]. The revised manuscript has the following sentence (and is limited to two self-citations as per Nutrients’ policy on self-citations):

Lines 94-96:  The sample size and 8 week duration are based on improvement of metabolic parameters with other dietary interventions in Zucker rats [1,2].

  1. Why was the experiment period chosen to be eight weeks? Is there any theoretical basis for the above?

Response:  The 8 weeks experimental period was based on our previous experience with dietary interventions in fa/fa Zucker rats and for which there were improvements in oral glucose tolerance, pancreatic islets, hepatic steatosis, and other metabolic parameter in a 8 week study [2-7].  The revised manuscript has the following sentence (and is limited to two self-citations as per Nutrients’ policy on self-citations):

Lines 94-96:  The sample size and 8 week duration are based on improvement of metabolic parameters with other dietary interventions in Zucker rats [1,2].

  1. Dosage of zinc supplementation: The zinc supplementation dose used in the study (300 mg Zn/kg diet) varied significantly from the zinc control dose (30 mg Zn/kg diet) to the marginal deficiency dose (4 mg Zn/kg diet). Consideration needs to be given to whether such high-dose supplementation is clinically necessary or whether there is a risk of excessive intake. How to determine the best dose supplement?

Response:  This research was designed as a preclinical study to investigate the effects of marginal zinc deficiency and zinc supplementation on various metabolic parameters in ZDF rats. We had unpublished data indicating that 150 mg Zn/kg diet for 8 weeks had minimal effects for increasing zinc in femurs and other tissues of obese fa/fa Zucker rats compared to what might be expected for lean rats. We also knew from a previous study that 300 mg Zn/kg diet increased pancreatic zinc concentrations in obese db/db mice, but not lean mice, compared to db/db and lean mice receiving 30 mg Zn/kg diet, respectively [8]. Thus, for the present experiment we chose 300 mg Zn/kg diet for the zinc supplementation and we included parameters such as liver copper and liver iron to determine if there are effects on other minerals in an obese model. Since this is a preclinical study in an obese rat model (with a genetic defect in the leptin receptor), it is inappropriate to extrapolate the dose to what is clinically necessary for humans. Various meta-analyses are evaluating the zinc dose required for improving glycemia or lipid parameters in individuals with type 2 diabetes [9-13] . Unlike human studies, this preclinical study makes it possible to investigate the relationships among dietary zinc intake, the levels of zinc in key tissues, and the effects of metabolic parameters.

  1. In the selection of zinc supplement dosage, have you considered the effects of long-term supplementation on animal health?

Response:  The zinc supplement dosage of 300 mg Zn/kg diet was selected for testing the effects on key metabolic parameters in preclinical study of 8 week duration. The results demonstrating that reductions in liver copper and liver iron (due to competition with intestinal absorption of zinc, copper and iron), indicate that there are concerns about the effects of long-term supplementation on animal health if ZDF rats were to consume this level of zinc long-term.  In the Discussion, lines 1496-1504, we address how “the present study also reinforces that there is a very narrow range for zinc supplementation without negative effects on other divalent metal ions [14] as the ZS ZDF rats had a reduction in liver copper concentration and total liver copper compared to ZC ZDF.”

  1. ZDF rats with borderline zinc deficiency (MZD) reported reduced zinc concentrations in the femur and pancreas, but no changes in serum zinc or body weight. This may suggest that assessing zinc status by serum zinc concentration alone may not be accurate enough. More biomarkers need to be considered to more fully assess zinc nutritional status.

Response:  Thank you for this point.  We agree with the reviewer that multiple biomarkers need to be considered to fully assess zinc nutritional status and particularly with a marginal (or borderline) zinc deficiency. In the Discussion, lines 1436-1438, we point out that “In the present study, the ZDF rats with marginal zinc deficiency did not have hypozincemia despite a 50% lower femur zinc concentration and a 30% lower pancreatic zinc concentration compared to both ZC ZDF and ZC lean rats.” This reinforces other observations in the literature that serum zinc alone is not a reliable marker of for marginal or borderline zinc deficiency.

  1. The potential role of zinc in diabetes management is discussed, but its specific guiding implications for clinical practice, including recommended dosage and monitoring requirements for zinc supplementation, are not specified.

Response:  It is not appropriate to develop specific guiding implications for clinical practice from this preclinical study in an obese rat model. With respect to monitoring requirements for zinc supplementation in humans, clinical studies are limited to assessments of molecules circulating in the blood. Based on our results that serum zinc does not reflect differences in pancreatic or femur zinc in a preclinical model, we have suggested in the last line of the following excerpt that a method (or a biomarker) to non-invasively assess femur or pancreatic zinc concentrations would be very helpful for advancing our understanding of zinc and diabetes in humans:

Lines 1434-1443:  The link between lower zinc status in humans and diabetes is based on studies linking reduced dietary zinc intake or serum zinc concentrations with risk for diabetes or presence of disease [15-17]. In the present study, the ZDF rats with marginal zinc deficiency did not have hypozincemia despite a 50% lower femur zinc concentration and a 30% lower pancreatic zinc concentration compared to both ZC ZDF and ZC lean rats. It has been notoriously difficult to assess mild zinc deficiency in humans based on circulating zinc concentrations [14,18,19] and the present study raises questions about whether the availability of a method (or a biomarker) to non-invasively assess femur or pancreatic zinc concentrations in humans would help advance our understanding of how reduced zinc status impacts the risk and/or progression of type 2 diabetes in humans.

References

  1. Stringer, D.M.; Zahradka, P.; Declercq, V.C.; Ryz, N.R.; Diakiw, R.; Burr, L.L.; Xie, X.; Taylor, C.G. Modulation of lipid droplet size and lipid droplet proteins by trans-10,cis-12 conjugated linoleic acid parallels improvements in hepatic steatosis in obese, insulin-resistant rats. Biochimica et biophysica acta 2010, 1801, 1375-1385, doi:10.1016/j.bbalip.2010.08.011.
  2. Noto, A.; Zahradka, P.; Ryz, N.R.; Yurkova, N.; Xie, X.; Taylor, C.G. Dietary conjugated linoleic acid preserves pancreatic function and reduces inflammatory markers in obese, insulin-resistant rats. Metabolism: clinical and experimental 2007, 56, 142-151, doi:10.1016/j.metabol.2006.09.009.
  3. Noto, A.; Zahradka, P.; Yurkova, N.; Xie, X.; Nitschmann, E.; Ogborn, M.; Taylor, C.G. Conjugated linoleic acid reduces hepatic steatosis, improves liver function, and favorably modifies lipid metabolism in obese insulin-resistant rats. Lipids 2006, 41, 179-188, doi:10.1007/s11745-006-5086-6.
  4. Noto, A.; Zahradka, P.; Yurkova, N.; Xie, X.; Truong, H.; Nitschmann, E.; Ogborn, M.R.; Taylor, C.G. Dietary conjugated linoleic acid decreases adipocyte size and favorably modifies adipokine status and insulin sensitivity in obese, insulin-resistant rats. Metabolism: clinical and experimental 2007, 56, 1601-1611, doi:10.1016/j.metabol.2007.06.025.
  5. Hong, L.; Zahradka, P.; Cordero-Monroy, L.; Wright, B.; Taylor, C.G. Dietary Docosahexaenoic Acid (DHA) and Eicosapentaenoic Acid (EPA) Operate by Different Mechanisms to Modulate Hepatic Steatosis and Hyperinsulemia in fa/fa Zucker Rats. Nutrients 2019, 11, doi:10.3390/nu11040917.
  6. Hong, L.; Zahradka, P.; Taylor, C.G. Differential Modulation by Eicosapentaenoic Acid (EPA) and Docosahexaenoic Acid (DHA) of Mesenteric Fat and Macrophages and T Cells in Adipose Tissue of Obese fa/fa Zucker Rats. Nutrients 2024, 16, doi:10.3390/nu16091311.
  7. Wojcik, J.L.; Devassy, J.G.; Wu, Y.; Zahradka, P.; Taylor, C.G.; Aukema, H.M. Protein source in a high-protein diet modulates reductions in insulin resistance and hepatic steatosis in fa/fa Zucker rats. Obesity (Silver Spring, Md.) 2016, 24, 123-131, doi:10.1002/oby.21312.
  8. Simon, S.F.; Taylor, C.G. Dietary zinc supplementation attenuates hyperglycemia in db/db mice. Experimental biology and medicine (Maywood, N.J.) 2001, 226, 43-51, doi:10.1177/153537020122600107.
  9. Wang, X.; Wu, W.; Zheng, W.; Fang, X.; Chen, L.; Rink, L.; Min, J.; Wang, F. Zinc supplementation improves glycemic control for diabetes prevention and management: a systematic review and meta-analysis of randomized controlled trials. The American journal of clinical nutrition 2019, 110, 76-90, doi:10.1093/ajcn/nqz041.
  10. Asbaghi, O.; Sadeghian, M.; Fouladvand, F.; Panahande, B.; Nasiri, M.; Khodadost, M.; Shokri, A.; Pirouzi, A.; Sadeghi, O. Effects of zinc supplementation on lipid profile in patients with type 2 diabetes mellitus: A systematic review and meta-analysis of randomized controlled trials. Nutrition, metabolism, and cardiovascular diseases : NMCD 2020, 30, 1260-1271, doi:10.1016/j.numecd.2020.03.021.
  11. Nazari, M.; Nikbaf-Shandiz, M.; Pashayee-Khamene, F.; Bagheri, R.; Goudarzi, K.; Hosseinnia, N.V.; Dolatshahi, S.; Omran, H.S.; Amirani, N.; Ashtary-Larky, D., et al. Zinc Supplementation in Individuals with Prediabetes and type 2 Diabetes: a GRADE-Assessed Systematic Review and Dose-Response Meta-analysis. Biological trace element research 2023, 10.1007/s12011-023-03895-7, doi:10.1007/s12011-023-03895-7.
  12. Ghaedi, K.; Ghasempour, D.; Jowshan, M.; Zheng, M.; Ghobadi, S.; Jafari, A. Effect of zinc supplementation in the management of type 2 diabetes: A grading of recommendations assessment, development, and evaluation-assessed, dose-response meta-analysis of randomized controlled trials. Critical reviews in food science and nutrition 2023, 10.1080/10408398.2023.2209802, 1-12, doi:10.1080/10408398.2023.2209802.
  13. Heidari Seyedmahalleh, M.; Montazer, M.; Ebrahimpour-Koujan, S.; Azadbakht, L. The Effect of Zinc Supplementation on Lipid Profiles in Patients with Type 2 Diabetes Mellitus: A Systematic Review and Dose-Response Meta-Analysis of Randomized Clinical Trials. Advances in nutrition (Bethesda, Md.) 2023, 14, 1374-1388, doi:10.1016/j.advnut.2023.08.006.
  14. Maret, W.; Sandstead, H.H. Zinc requirements and the risks and benefits of zinc supplementation. Journal of trace elements in medicine and biology : organ of the Society for Minerals and Trace Elements (GMS) 2006, 20, 3-18, doi:10.1016/j.jtemb.2006.01.006.
  15. Chausmer, A.B. Zinc, insulin and diabetes. Journal of the American College of Nutrition 1998, 17, 109-115, doi:10.1080/07315724.1998.10718735.
  16. de Carvalho, G.B.; Brandão-Lima, P.N.; Maia, C.S.; Barbosa, K.B.; Pires, L.V. Zinc's role in the glycemic control of patients with type 2 diabetes: a systematic review. Biometals : an international journal on the role of metal ions in biology, biochemistry, and medicine 2017, 30, 151-162, doi:10.1007/s10534-017-9996-y.
  17. Jansen, J.; Karges, W.; Rink, L. Zinc and diabetes--clinical links and molecular mechanisms. The Journal of nutritional biochemistry 2009, 20, 399-417, doi:10.1016/j.jnutbio.2009.01.009.
  18. Hennigar, S.R.; Lieberman, H.R.; Fulgoni, V.L., 3rd; McClung, J.P. Serum Zinc Concentrations in the US Population Are Related to Sex, Age, and Time of Blood Draw but Not Dietary or Supplemental Zinc. The Journal of nutrition 2018, 148, 1341-1351, doi:10.1093/jn/nxy105.
  19. Hall, A.G.; King, J.C. Zinc Fortification: Current Trends and Strategies. Nutrients 2022, 14, doi:10.3390/nu14193895.

Reviewer 2 Report

Comments and Suggestions for Authors

This paper indicated that differences in zinc levels ingested affect greatly the obesity, glycemic control, pancreatic islets, hepatic steatosis and renal function in Zucker diabetic fatty (ZDF) rats, by using both marginal zinc deficiency and zinc supplementation. Overall results and discussions obtained from this study are interesting and valuable for relationships between diabetes mellitus and zinc state in body, and thus deemed worthy of publication in Nutrients. 
However, b
efore the acceptance, the authors need to resolve the following questions.

1. Throughout the paper, there are multiple places where GPR-39 is incorrectly written as GRP-39. Please correct all of these, including the incorrect notations in the figures.

2. Why are all data presented as the means ±
standard errors rather than means ± standard deviations?  For the present experimental design, the data should be denoted by means ± standard deviation.

3. In Figure 1 B, the unit on the vertical axis (µmol/L) for femur zinc level clearly appears to be incorrect because the zinc concentrations are per organ weight in femur (bone).

4. In both Scheme 1 A and Scheme 2 A, why are there two photographs of Western blotting, one on the left and one on the right?  Additionally, the quality for all photographs of Western blotting is very poor. Could you replace or change to another representative result?

Author Response

Thank you for the review of our manuscript, pointing out the errors, and your insightful comments for improving our manuscript.

This paper indicated that differences in zinc levels ingested affect greatly the obesity, glycemic control, pancreatic islets, hepatic steatosis and renal function in Zucker diabetic fatty (ZDF) rats, by using both marginal zinc deficiency and zinc supplementation. Overall results and discussions obtained from this study are interesting and valuable for relationships between diabetes mellitus and zinc state in body, and thus deemed worthy of publication in Nutrients
However, before the acceptance, the authors need to resolve the following questions.

1. Throughout the paper, there are multiple places where GPR-39 is incorrectly written as GRP-39. Please correct all of these, including the incorrect notations in the figures.

Response:  Thank you for noticing our error.  The text and figure have been corrected to GPR-39.

2. Why are all data presented as the means ± standard errors rather than means ± standard deviations?  For the present experimental design, the data should be denoted by means ± standard deviation.

Response:  All the figures and tables have been updated to present the data as means ± standard deviations.

  1. In Figure 1 B, the unit on the vertical axis (µmol/L) for femur zinc level clearly appears to be incorrect because the zinc concentrations are per organ weight in femur (bone).

Response:  Thank you for noticing our error. The vertical axis for Figure 1B has been corrected to Femur Zinc (µmol/g dry weight).

  1. In both Scheme 1 A and Scheme 2 A, why are there two photographs of Western blotting, one on the left and one on the right?  Additionally, the quality for all photographs of Western blotting is very poor. Could you replace or change to another representative result?

Response:  The titles for these figures have been corrected to Supplementary Figure 1 and Supplementary Figure 2. The two images for Western blotting are provided to show the results for n=2 animals/group. The complete images for Western blotting have been provided to Nutrients. These blots were developed on film and we do not have another version of the film development. We agree that the quality of the images is not good, however, the results are representative for no changes in protein levels of GPR-39, ZnT3, and MCP-1, and lower levels of SOCS3 and TNF-alpha (precursor and mature) in ZDF rats versus lean Zucker rats.